# Prevention of *Herpesviridae* Infections by Cationic PEGylated Carbosilane Dendrimers

**DOI:** 10.3390/pharmaceutics14030536

**Published:** 2022-02-28

**Authors:** Elena Royo-Rubio, Vanessa Martín-Cañadilla, Marco Rusnati, Maria Milanesi, Tania Lozano-Cruz, Rafael Gómez, José Luís Jiménez, Maria Ángeles Muñoz-Fernández

**Affiliations:** 1Laboratorio InmunoBiologia Molecular, Instituto Investigacion Sanitaria Gregorio Maranon (IiSGM), Hospital General Universitario Gregorio Maranon (HGUGM), 28009 Madrid, Spain; elena.royo@iisgm.com (E.R.-R.); vanessa.canadilla@iisgm.com (V.M.-C.); 2Plataforma de Laboratorio (Inmunologia), HGUGM, IiSGM, Spanish HIV HGM BioBank, 28009 Madrid, Spain; jjimenezf.hgugm@salud.madrid.org; 3Department of Molecular and Translational Medicine, University of Brescia, 25123 Brescia, Italy; marco.rusnati@unibs.it (M.R.); m.milanesi006@unibs.it (M.M.); 4Departmento Quimica Organica y Quimica Inorganica, Instituto de Investigacion Quimica “Andres M. del Rio″ (IQAR), Universidad de Alcalá (IRYCIS), Campus Universitario, 28871 Madrid, Spain; tania.lozano@uah.es (T.L.-C.); rafael.gomez@uah.es (R.G.); 5Networking Research Center on Bioengineering, Biomaterials and Nanomedicine (CIBER-BBN), 28029 Madrid, Spain

**Keywords:** *Herpesviridae*, cationic dendrimers, HSPG, VHS-2 infection, HCMV infection, nanotechnology, inhibition

## Abstract

Infections caused by viruses from the *Herpesviridae* family produce some of the most prevalent transmitted diseases in the world, constituting a serious global public health issue. Some of the virus properties such as latency and the appearance of resistance to antiviral treatments complicate the development of effective therapies capable of facing the infection. In this context, dendrimers present themselves as promising alternatives to current treatments. In this study, we propose the use of PEGylated cationic carbosilane dendrimers as inhibitors of herpes simplex virus 2 (HSV-2) and human cytomegalovirus (HCMV)infections. Studies of mitochondrial toxicity, membrane integrity, internalization and viral infection inhibition indicated that G2-SN15-PEG, G3-SN31-PEG, G2-SN15-PEG fluorescein isothiocyanate (FITC) labeled and G3-SN31-PEG-FITC dendrimers are valid candidates to target HSV-2 and HCMV infections since they are biocompatible, can be effectively internalized and are able to significantly inhibit both infections. Later studies (including viral inactivation, binding inhibition, heparan sulphate proteoglycans (HSPG)binding and surface plasmon resonance assays) confirmed that inhibition takes place at first infection stages. More precisely, these studies established that their attachment to cell membrane heparan sulphate proteoglycans impede the interaction between viral glycoproteins and these cell receptors, thus preventing infection. Altogether, our research confirmed the high capacity of these PEGylated carbosilane dendrimers to prevent HSV-2 and HCMV infections, making them valid candidates as antiviral agents against *Herpesviridae* infections.

## 1. Introduction

*Herpesviridae* is a large family of enveloped double-stranded DNA viruses that comprises a wide variety of herpesviruses with different biological characteristics, but that have in common basic properties such as their ability to infect animals (including humans), their morphology, their genetic complexity and a high regulated transcription [1]. 

The International Committee on Taxonomy of Viruses (ICTV) established the division of this family in three subfamilies: (i) *Alfaherpesviridae*, (ii) *Gammaherpesviridae,* and (iii) *Betaherpesviridae* [2]. In turn, they are divided into different subtypes, including herpes simplex virus 1 and 2 (HSV-1 and HSV-2), Epstein–Barr virus (EBV), varicella zoster virus (VZV), or human cytomegalovirus (HCMV). Those are among the most widespread pathogens in the word; in fact, more than 90% of adults have been infected with one of these subtypes [3,4]. These viruses are able to infect a wide range of cells and the pathogenesis of the infection can range from mild lytic infections to severe persistent and latent infections with many recurrences, and there is still no cure [5]. 

One of the most prevalent infections is the one produced by HSV-2 which, according with the latest available data, affects nearly 500 million people, mainly women from Africa, aged between 15 to 49 years old [6]. This infection, almost entirely sexually transmitted, produces symptoms such as ulcers on the external genitalia and perineum, dysuria, and painful inguinal lymphadenopathy. Other rare complications caused by HSV-2 infection are lip lesions, herpetic hepatitis, or aseptic meningitis [7]. Another concerning infection is the one produced by HCMV; transmission of this virus may occur transplacentally, through breastfeeding, by intimate contact or transplantation [8]. Congenital infection is estimated in between 0.7 and 5% of all births, and is the leading cause of neurological impairment in infants, including hearing and vision loss, microcephaly, or developmental and motor delay. In adults, HCMV is mainly asymptomatic but can lead to life-threatening diseases in immunocompromised individuals, producing manifestations such as mononucleosis-like syndrome, tissue-invasive disease, neutropenia, interstitial pneumonia or autoimmune phenomena (vasculitis, scleroderma, systemic lupus erythematosus, etc.) [9]. 

Virions of the *Herpesviridae* family have four main shared structural components: a core in which the dsDNA is wrapped, encased in an icosahedral capsid composed of 12 pentameric and 150 hexameric capsomers, coated with a tegument formed by variable amounts of globular material and an envelope surrounding the structure formed by different proteins and glycoproteins in a lipidic bilayer [10]. Initial contact between viral envelope glycoproteins (mainly gB, gH and gL) and cell membrane takes place usually through different types of cell proteoglycans, usually heparan sulphate proteoglycans (HSPGs) [11]. This occurs between the negatively charged sulphated groups of the heparin-like glycosaminoglycan (GAG) chains of HSPGs and stretches of basic amino acids (referred as heparin-binding domains) present within the viral glycoproteins [11,12,13]. This first association with HSPGs usually favours the following binding of the virus to its specific entry receptor and hence infection. Herpesviruses have extensive cell tropism, with a wide variety of entry routes that depend on viral determinants and cell types; however, in all cases, these interactions trigger conformational changes which lead to membrane fusion and subsequent viral entry [14]. 

Currently, there is no treatment that completely clearances HSV-2 and HCMV, mainly due to the capacity of these viruses to establish latent infections in the host and to the appearance of resistance against these drugs. Ongoing therapies for HSV-2 infections are based on nucleoside analogues to inhibit DNA polymerases, such as acyclovir, valacyclovir, or famciclovir in order to control viral replication, disease progression, recurrences and transmission [15]. Related to HCMV, infants with congenital CMV infection and people with immunodeficiencies are mostly treated with ganciclovir or its pro-drug valganciclovir, other inhibitors of DNA polymerases based on nucleoside analogues [16]. These treatments have proved effective in the prevention and treatment of different CMV infections, including retinitis, gastrointestinal manifestations, pneumonia, polyradiculopathy or mononeuritis [17].

The absence of a definitive therapy, the fact that herpesviruses have a broad cell tropism, and the appearance of resistances against existing treatments makes it mandatory to develop new therapeutic approaches to face the present situation. Our group has carried out broad research focused on the use of nanotechnology against viral infections [18,19,20,21]. Specifically, these works are based on the use of dendrimers, which are three-dimensional hyperbranched molecules formed by a nuclear core around which branched units (named dendrons) are built; these dendrons present diverse functional groups on their periphery, which endow unique physicochemical characteristics [22,23]. These molecules are characterized by a controlled synthesis, high biocompatibility, low polydispersity or polyvalency which make them better instruments than other polymers to face viral infections [24,25]. In this work, we present the use of four PEGylated cationic carbosilane dendrimers (PCCDs), G2-SN15-PEG, G3-SN31-PEG, G2-SN15-PEG-FITC and G3-SN31-PEG-FITC, which are characterized by the presence of positively charged functional terminal groups and PEGylation residues [26]. These two features make them great candidates to be used against *Herpesviridae* infections due to their improved biocompatibility and the positive charges at the periphery which can bind to HSPGs, thus masking these receptors and preventing further viral infections [27]. Hence, the objective of this work is to develop a new promising therapy against HSV-2 and HCMV infections based on the use of PCCDs as inhibitors of the interaction between viral glycoproteins and cell HSPGs. 

## 2. Materials and Methods

### 2.1. Cell Lines 

The Vero cell line, a fibroblast-like kidney cell from the African green monkey, was obtained from the American Type Culture Collection (ATCC) (CCL-81^TM^, ATCC, Manassas, VA, USA). Cells were cultured in Dulbecco’s modified Eagle’s medium (DMEM) (Biochrom GmbH, Berlin, Germany) supplemented with 5% heat-inactivated fetal bovine serum (FBS) (Biochrom GmbH, Berlin, Germany), 2 mM L-glutamine (Lonza, Base, Switzerland), and a cocktail of antibiotics formed by 125 mg/mL ampicillin, 125 mg/mL cloxacillin and 40 mg/mL gentamicin (Normon, Madrid, Spain). 

The MRC-5 cell line, a human fibroblast lung cell, was obtained from ATCC (CCL-171^TM^, ATCC, Manassas, VA, USA). This cell line was maintained in Eagle’s minimum essential medium (EMEM) (ATCC, Manassas, VA, USA) supplemented with 10% FBS, 2 mM L-glutamine, and the previously mentioned cocktail of antibiotics. Both cell lines were cultured at 37 °C with 5% CO_2_.

### 2.2. Viral Isolates

Viral strain HSV-2_333_ (GenBank accession number LS480640, NIH, Bethesda, MD, USA) and HCMV_AD-169_ (ATCC VR-538^TM^, ATCC, Manassas, VA, USA) were grown and propagated in Vero and MRC-5 cells, respectively, and titrated by plaque assay with serial dilutions as previously described [28,29]. After ultracentrifugation, stock aliquots were stored at −80 °C.

### 2.3. Dendrimers and Reagents

PEGylated cationic carbosilane dendrimers (PCCDs)—G2-SN15-PEG and G3-SN31-PEG—and their FITC-labelled forms were synthesized according to methods reported by Dendrimers for Biomedical Applications Group of the University of Alcalá (Alcalá de Henares, Madrid, Spain) [26]. Stock solutions of dendrimers (1 mM) and their subsequent dilutions were prepared with nuclease-free water (Promega, Madrid, Spain). Table 1 includes a brief description of these dendrimers.

Acliclovir 50 mg (Selleckchem, Houston, TX, USA) was purchased as a lyophilised product and reconstituted into a 10 mM dilution with dimethyl sulfoxide (DMSO, Honeywell, Charlotte, NC, USA). Ganciclovir 500 mg (Hoffmann-La Roche, Basel, Switzerland) was also purchased as a lyophilised product and reconstituted into a 5.4 mg/mL dilution with nuclease-free water (Promega, Madrid, Spain).

Heparin (13.6 kDa) was obtained from Laboratori Derivati Organici Spa (Milan, Italy). Heparinase II from *Flavobacterium heparinum* was obtained from Merck KGaA (Merck Group, Darmstadt, Germany). 

### 2.4. Mitochondrial Activity Assay

Evaluation of the dendrimer-induced mitochondrial toxicity was done by the MTT assay (Sigma, St Louis, MO, USA). This assay, based on the reduction of the 3-(4-5-dimethylthiazol-2-yl)-2,5-diphenyltetrazolium bromide (MTT) into formazan crystals, allows determination of the cellular mitochondrial metabolism. Briefly, following manufacturer’s instructions, Vero and MRC-5 cells were seeded at a density of 1.5 × 10^4^ cells/well into 96-well plates. After incubating cells for 24 h at 37 °C, cells were treated with different concentrations of PCCDs for 48 h and 6 days, respectively, to ensure that this treatment would not affect viability during inhibition experiments. After incubation, the medium was discarded and a solution formed by MTT (5 mg/mL) and Opti-MEM^TM^ (Thermo Fisher Scientific, Waltham, MA, USA) (1:11) was added. Two hours later, the reaction was stopped by removing the solution and dissolving formazan crystals in DMSO (Honeywell, Charlotte, NC, USA). Absorbance was recorded in a Synergy 4 plate reader (BioTek, Winooski, VT, USA) at 490 nm. Culture medium was used as non-treated control and DMSO 10% as cellular death control. Measurements were performed in triplicate.

### 2.5. Membrane Integrity Assay 

Determination of cellular toxicity was done by the lactate dehydrogenase (LDH) CytoTox 96^®^ Non-Radioactive Cytotoxicity assay (Promega, Spain, Madrid). Shortly, following manufacturer’s instructions, both cell lines were seeded at a density of 1.5 × 10^4^ cells/well into 96-well plates. Then, 24 h after, cells were treated with increasing concentrations of PCCDs for 48 h (Vero cells) and 6 days (MRC-5 cells), again to ensure no toxicity during inhibition experiments. Afterwards, cells were lysed for 45 min at 37 °C in 0.9% Triton X-100 (Promega, Madrid, Spain). Then, 50 µL of LDH reagent (Promega, Spain, Madrid) were added and incubated in the dark for 30 min at room temperature. Absorbance was recorded in a Synergy 4 plate reader at 490 nm. The culture medium was used as a non-treated control. Measurements were performed in triplicate.

### 2.6. Confocal Microscopy 

Internalization of cationic dendrimers into Vero and MRC-5 cells was studied by confocal microscopy using a Leica TSC SPE Confocal Microscope (Leica, Wetzalar, Germany). Vero and MRC-5 cells were seeded at a density of 1.75 × 10^5^ and 8 × 10^4^ cells, respectively, in 12 mm circle cover slips (Thermo Fisher Scientific, Waltham, MA, USA) pre-treated with poly-L-Lysine (Sigma, St Louis, MO, USA). Cells were treated with the maximum non-toxic concentrations of FITC-labelled PCCDs (G2-SN15-PEG FITC and G3-SN31-PEG FITC) for 2 h, 6 h and 24 h at 37 °C. After incubation, cells were rinsed with 3% bovine serum albumin (BSA, Sigma, St Louis, MO, USA) phosphate buffered saline (PBS, Lonza, Base, Switzerland). Cell fixation was performed with 4% paraformaldehyde (PFA, Panrea, Barcelona, Spain) for 15 min and permeabilization with 0.1% Triton 100X (Sigma, St Louis, MO, USA) for 15 min. Actin labelling was carried out by incubating cells with Alexa Fluor^®^ 555 Phalloidin (Thermo Fisher Scientific, Waltham, MA, USA) for 1 h at room temperature (RT). Following two rinses with 3% BSA PBS, cells were incubated with 4′,6-Diamidino-2-phenylindole dihydrochloride (DAPI, Sigma, St Louis, MO, USA) for nuclear visualization. Lastly, cells were mounted in microscope slides (Dako, Carpinteria, CA, USA) with fluorescent mounting media (Dako, Carpinteria, CA, USA). ImageJ (National Institutes of Health, Bethesda, MD, USA) was used to analyse the images.

### 2.7. Flow Cytometry

Flow cytometry was used to confirm the internalization of FITC labelled dendrimers into both cell lines. Briefly, Vero and MRC-5 cells were seeded at a density of 1.75 × 10^5^ and 6 × 10^4^ cells, respectively, in 24 well-plates and treated with 1 µM G2-SN15-PEG FITC and 0.5 µM G3-SN31-PEG FITC for 1 h, 2 h, 6 h and 24 h. Viable cells were identified using LIVE/DEAD™ Fixable Aqua Dead Cell Stain (Thermo Fisher Scientific, Waltham, MA, USA). Lastly, cells were fixed with 3% paraformaldehyde. Flow cytometry was performed on a Galios (Beckman Coulter, Brea, CA, USA). Kaluza software 2.1 (Beckman Coulter, Brea, CA, USA) was used for the analysis of the measurements. 

### 2.8. Inhibition Assay 

The antiviral activity of the different dendrimers was evaluated performing inhibition experiments by plaque reduction assay. In-brief, Vero and MRC-5 cells were seeded at a density of 1.75 × 10^5^ and 6 × 10^4^ cells/well in 24-well plates, respectively, and incubated at 37 °C for 24 h. Both cell lines were then treated with increasing concentrations from 0.2 µM to the maximum non-toxic of G2-SN15-PEG and G3-SN31-PEG dendrimers for 1 h. In addition, increasing concentrations (from 0.2 µM to 10 µM) of reference treatments Acyclovir or Ganciclovir were used for comparison with current reference treatments. Afterwards, pre-treated Vero and MRC-5 cells were infected with the viral strains HSV-2_333_ and HCMV_AD-169_, respectively, at a multiplicity of infection (MOI) of 0.001. Three hours after infection, cells were washed with PBS to remove unabsorbed viruses. HSV-2 infection remained in DMEM supplemented with 2% FBS and 0.4% IgG (Berigloblin P, CSL Behring, King of Prussia, PA, USA) for 48 h. HCMV infection remained in EMEM supplemented with 2% FBS for 6 days. Then, medium was removed and both cell lines were stained with 300 mg/L Methylene Blue (Sigma, St Louis, MO, USA) for 30 min (Vero) and 3 h (MRC-5). Inhibition was determined as the reduction of the plaques formed with treatments regarding the infection control. Non-infected and non-treated samples were used as non-treated (NT) and infection controls (IC), respectively. 

### 2.9. Viral Inactivation Assay

To determine if the observed inhibition is a result of the direct interaction of PCCDs with HSV-2 or HCMV, a viral inactivation assay was performed. Briefly, Vero and MRC-5 cells were seeded at a density of 1.75 × 10^5^ and 6 × 10^4^ cells/well in 24-well plates, respectively, and incubated at 37 °C for 24 h. Then, G2-SN15-PEG or G3-SN31-PEG dendrimers at 1 µM were incubated with 175 pfu/mL or 60 pfu/mL of cell free HSV-2_333_ and HCMV_AD-169_, respectively, for 2 h at 37 °C. After incubation, the mixture was centrifuged at 12,000 rpm for 1 h at 4 °C and the supernatant was discarded; then, the pellet was rinsed with PBS and centrifuged again at 12,000 rpm for 1 h at 4 °C. Afterwards, the supernatant was discarded and replaced either with fresh DMEM supplemented with 2% FBS and added to Vero cells, or fresh EMEM supplemented with 2% FBS and added to MRC-5 cells. Infections were revealed as previously described. Viral disruption positive control was obtained with Triton X-100 at 0.1% and a culture medium was used as a negative control. 

### 2.10. Binding Inhibition Assay

To resolve if the observed HSV-2 or HCMV inhibition is a result of the blockade of cell membrane viral-receptors by PCCDs, a binding inhibition study by plaque reduction assay was performed. Briefly, Vero and MRC-5 cells were seeded at a density of 1.75 × 10^5^ and 6 × 10^4^ cells/well in 24-well plates, respectively, and incubated at 37 °C for 24 h. Then, cells were pre-cooled at 4 °C for 15 min and treated with 1 µM G2-SN15-PEG or G3-SN31-PEG dendrimers for 1 h at 4 °C. Subsequently, cells were infected with 175 pfu/mL or 60 pfu/mL of HSV-2_333_ and HCMV_AD-169_, respectively, for 2 h at 4 °C. Afterwards, inoculum was discarded and replaced either with fresh DMEM supplemented with 2% FBS and 0.4% IgG in Vero cells, or fresh EMEM supplemented with 2% FBS in MRC-5 cells. Infections were revealed as previously described. Culture medium was used as a negative control.

### 2.11. HSPG Binding Assay

Heparin competition assay was carried out to study the interactions between PCCDs and HSPGs. To do so, cells were seeded in 24-well plates and incubated for 2 h at 4 °C in PBS with 1 µM G2-SN15-PEG or 0.5 µM G3-SN31-PEG dendrimers in the absence or presence of 10 µg/mL of heparin, a structurally similar molecule [30]. In additional experiments, cells were incubated for 1 h at 37 °C with dendrimers at the same concentration and washed with PBS containing 2.0 M NaCl, known to disrupt the binding to HSPGs [31]. Along with these experiments, cells were pre-treated with 200 mU/mL of heparinase II, acknowledged to inhibit HSPG-dependent binding [32], (Merck Group, Darmstadt, Germany) for 2 h at 37 °C, before incubating for 1 h at 37 °C with dendrimers at this concentration. Assessment of the amounts of cell-associated dendrimers after these studies was done by evaluating the mean fluorescence intensity (MFI) of dendrimers in cells by flow cytometry, as described in its corresponding section. 

### 2.12. Surface Plasmon Resonance 

Surface plasmon resonance (SPR) was conducted to study the capacity of dendrimers to bind to heparin/HSPGs. Measurements were performed on a BIAcore X100 instrument (Cytiva, Marlborough, MA, USA) using a research grade sensor chip SA (Cytiva, Marlborough, MA, USA) whose surface consists of a carboxy methylated dextran matrix pre-immobilized with streptavidin. One cell of the sensor chip was conditioned with three consecutive 1-min injections of 1.0 M NaCl in 50 mM NaOH; then, heparin biotinylated at its reducing end diluted in 10 mM HEPES buffer, pH 7.4, containing 150 mM NaCl, 3 mM EDTA, and 0.005% surfactant P20 (HBS-EP) was injected for 8 min at a flow rate of 10 µL/min, allowing the immobilization of 134.1 resonance units (RU) (equal to 9.8 fmol/mm^2^) of the GAG. Then, the heparin-containing flow cell was over-coated by injecting biotinylated BSA re-suspended in HBS-EP for 8 min at a flow rate of 10 µL/min, allowing immobilization of 104 RU (equal to 1.6 fmol/mm^2^) of the protein, expected to help mask the residual aspecific negatively charged binding site available on the surface. The remaining flow cell of the sensor chip was coated only with biotinylated BSA protein as described above, allowing the immobilization of 569.2 RU (equal to 8.6 fmol/mm^2^) of protein, and used to evaluate the nonspecific binding and for blank subtraction. For the study of their interaction with heparin, dendrimers G2-SN15-PEG and G3-SN31-PEG were re-suspended in HBS-EP and injected at increasing concentrations over the heparin and control BSA flow cells for 3 min (to allow their association with immobilized molecules) and then washed. After every run, the sensor chip was regenerated by injection of 2.0 M NaCl to detach the dendrimers that tend to remain bound to the biosensor surfaces. The dissociation constant (Kd, that is inversely proportional to the affinity binding) was calculated by fitting with the Scatchard’s equation for the plot of RU measured at equilibrium as a function of the ligand concentration in solution. All fitting was performed by a least-square minimization procedure based on the Levemburg–Marquardt algorithm. To evaluate the specificity of the binding of the compounds to surface-immobilized heparin, the compounds (0.75 μM) were injected onto the biosensor in the presence of a molar excess (5 mg/mL) of free heparin.

### 2.13. Statistics

The different statistical analyses were performed using GraphPad software Prism v.5.0 (GraphPad Software, San Diego, CA, USA). Data were obtained from two or three independent experiments performed by duplicate or triplicate. Data with two or three replicates are displayed as bars ± SD. A *p*-value of ≤ 0.05 was considered statistically significant (* *p* < 0.05; ** *p* < 0.005; *** *p* < 0.001).

## 3. Results

### 3.1. Cytotoxicity of the Dendrimers on Vero and MRC-5 Cell Lines

Evaluation of the cytotoxicity of the different dendrimers (G2-SN15-PEG, G3-SN31-PEG, G2-SN15-PEG FITC and G3-SN31-PEG FITC) in both cell lines was assessed by mitochondrial toxicity determination by the MTT assay and by membrane integrity by the LDH assay. Briefly, in both assays, Vero and MRC-5 cells were treated with increasing concentrations of dendrimers from 0.01 to 30 µM for 48 h and 6 days, respectively. Concentrations were considered non-toxic when the survival rate was ≥80%.

Results obtained from the MTT assay (Figure 1A,B) were more restrictive than those obtained from the LDH assay (Figure 1C,D), therefore we considered those results for the determination of the working concentrations. Non-toxic concentrations of the dendrimers for both cell lines are as follows: 1 µM, 1 µM, 1 µM and 0.5 µM for G2-SN15-PEG, G3-SN31-PEG, G2-SN15-PEG FITC and G3-SN31-PEG FITC, respectively. 

### 3.2. Internalization Study into Vero and MRC-5 Cells

We examined the capacity of FITC-labelled dendrimers to internalize into Vero and MRC-5 cell lines by confocal microscopy and flow cytometry. The sequence of the internalization process was studied by incubating both cell lines with either 1 µM G2-SN15-PEG FITC or 0.5 µM G3-SN31-PEG FITC dendrimers for 1 h, 2 h, 6 h or 24 h. To deepen its distribution, in confocal microscopy studies, actin filaments and the nucleus were labelled with phalloidin and DAPI, respectively. 

Entry analyses of both cell lines indicated that the G2-SN15-PEG FITC dendrimer has faster uptake dynamics than the G3-SN31-PEG FITC dendrimer, since it showed a significant increase of fluorescent positive cells after the first hour of incubation compared with the control (14% in Vero and 26% in MRC5) (Figure 2a and Figure 3a). In addition, the amount of positive fluorescent marks from both dendrimers showed an important increase along time points. The increment of signal in Vero cells from 1 h to 24 h from the G2-SN15-PEG FITC dendrimer was evident (45%), whilst for the G3-SN31-PEG FITC dendrimer it was not so remarkable (20%). In MRC-5 cells, both dendrimers showed a 30% increase in the number of positive cells from the first time point studied to the last. 

A more detailed study of the fluorescent distribution in both Vero and MRC-5 cells indicated that both dendrimers have a modified distribution pattern depending on the time elapsed from the initial distribution: short times of incubation showed peripheral distribution (co-localizing with F-actin), while longer times showed dendrimers to be in more internal regions with a punctate intracellular distribution (Figure 2b and Figure 3b). To summarize, both dendrimers can rapidly interact with the surface of both cell lines and be internalized into both Vero and MRC-5 cell lines, where they remain concentrated in a granular manner. 

### 3.3. Anti-HSV-2 and HCMV Activity of the Dendrimers

Inhibition experiments were performed to determine the antiviral activity of the non-labelled dendrimers front HSV-2 and HCMV infections. Vero and MRC-5 cell lines were treated with increasing concentrations of G2-SN15-PEG and G3-SN31-PEG dendrimers from 0.2 µM to the maximum non-toxic concentration (1 µM) one hour prior to infection with 0.001 MOI of HSV-2_333_ and HCMV_AD-169_. In addition, Acyclovir and Ganciclovir were used to compare our results with these reference treatments. After 48 h (Vero) and 6 days (MRC-5) of incubation, lysis plaques were revealed by methylene blue staining. 

Results shown in Figure 4a,b indicate that both G2-SN15-PEG and G3-SN31-PEG dendrimers are able to inhibit HSV-2 and HCMV infections at the maximum non-toxic concentrations. In both cases, the G3-SN31-PEG dendrimer presented better inhibition results at all tested concentrations, achieving values of 99% and 86% for HSV-2 and HCMV, respectively, at the maximum non-toxic concentration. Interestingly, both dendrimers present significantly better inhibition results than the actual reference treatments Acyclovir and Ganciclovir at their maximum non-toxic concentration. 

### 3.4. Cationic Dendrimers Prevent HSV-2 and HCMV at First Infection Stages

After demonstrating that both dendrimers have anti-HSV-2 and anti-HCMV activity, we proceeded to delve into the mechanism through which they carried out this activity. To do so, we studied two different mechanisms: viral inactivation (VI) and binding inhibition (BI). Briefly, for viral inactivation, dendrimers were incubated with cell-free HSV-2 or HCMV for 2 h at 37 °C. After two rounds of ultracentrifugation, the resuspended pellet was added to cells. For binding inhibition, cells were pre-cooled at 4 °C for 15 min and treated with dendrimers for 1 h at 4 °C, then cells were infected for 2 h at 4 °C. After 48 h (Vero) and 6 days (MRC-5) of incubation, lysis plaques were revealed by methylene blue staining. 

Figure 5a,b indicate that both dendrimers can significantly inhibit viral binding to both cell lines, obtaining similar inhibition values as the ones obtained in the antiviral activity study. On the other hand, G2-SN15-PEG and G3-SN31-PEG dendrimers are only able to inactivate HCMV and obtain lower inhibition values. These results suggest that inhibition is achieved at the first infection stages, most probably by impeding viral attachment to the cell membrane, thus preventing infection. 

### 3.5. Determination of Surface Membrane Interactions

Due to the cationic nature of PCCDs, we wondered if their inhibition potential depends on their capacity to bind to anionic HSPGs, hence preventing the binding of the virus to these receptors. To evaluate this possibility, different binding assays were carried out, starting with a competition assay with heparin, a structural analogue of HSPGs. Briefly, cells were treated with G2-SN15-PEG or G3-SN31-PEG dendrimers in the absence or presence of heparin. Alternatively, cells were treated with those dendrimers and washed with 2 M NaCl, a treatment known to detach basic proteins bound to heparin/HSPGs [27]. In other experiments, cells were pre-treated with heparinase II before incubation with dendrimers. After performing these assays, either the percentage of fluorescence positive cells or the mean fluorescence intensity (MFI) of the dendrimers was measured to infer the remaining amounts of cell-associated dendrimers. 

Results shown in Figure 6a,b demonstrate that HSPGs play a prominent role in the interaction of PCCDs with the cell surface. This can be observed in the remarkable drop in the number of cell-associated dendrimers (both G2-SN15-PEG FITC and G3-SN31-PEG FITC dendrimers) in both cell lines after treatment with heparin. These interactions were also partially disrupted when washing with 2 M NaCl, but, despite a remarkable reduction of the number of cell-associated dendrimers in all samples, differences were only significant for the G2-SN15-PEG FITC dendrimer in the Vero cell line and for the G3-SN31-PEG FITC dendrimer in the MRC-5 cell line. In the heparinase assay, Figure 6c,d indicates a notable decrease of binding capacity of dendrimers after treatment with heparinase II, observed in the notorious decline of the MFI in every sample; however, this decrease was only statistically significant for the G2-SN15-PEG FITC in the Vero cell line.

Due to the strong structural analogy existing between heparin and HSPGs, a surface plasmon resonance (SPR) biosensor containing immobilized heparin represents a simplified “cell-free” model that resembles the interaction of proteins or synthetic compounds to cell-associated HSPGs in vivo [33] that we have thus exploited to additionally evaluate the heparin/HSPGs binding capacity of PCCDs. As shown in Figure 7a,b, despite a significant aspecific binding to the BSA-containing surface (here used as a negative control), both dendrimers showed a significant specific binding to surface-immobilized heparin. The overlay of blank-subtracted sensorgrams obtained by injecting the dendrimers at increasing concentrations onto the sensor chip demonstrated a dose-response binding. Additionally, it is apparent that the binding of the dendrimers to heparin is very stable since they do not detach spontaneously from heparin at the end of the injection phase and can be removed only by a high salt washing (Figure 7c,d). From these sensorgrams, the values of binding at equilibrium of the dendrimers were used to obtain the dose-response curves shown in Figure 7e,f, and to calculate Kd values. Both G2-SN15-PEG and G3-SN31-PEG dendrimers bind surface-immobilized heparin in a saturable manner with a relatively high affinity (Kd in the high nanomolar, Table 2). The specificity of the binding is demonstrated by the fact that a molar excess of heparin completely abolishes the binding of the dendrimers to surface-immobilized heparin (Figure 7e,f).

## 4. Discussion

Years of research on the host entry mechanism of viruses from the *Herpesviridae* family has led to the conclusion that, despite the extensive cell tropism that they exhibit, host cell infection takes place through a conserved mechanism [4]. The entry machinery is formed by a conserved core fusion machinery (formed the heterodimer gH-gL and the fusion protein gB) and by divergent proteins such as gD for HSV-2 or gO for HCMV [34,35]. The latter act as ligands that bind to different host cell receptors, being the first step of the entry process the interaction with distinct classes of HSPGs located on the exterior surface of target cells [36]. This critical role makes HS a therapeutic target to develop novel inhibitors of HS-mediated binding and subsequent infection of herpesviruses. 

In this sense, we have studied the therapeutic effect of the use of the novel G2-SN15-PEG and G3-SN31-PEG dendrimers to prevent HSV-2 and HCMV infections based on their specific structures. Two characteristics of the periphery of these PCCDs makes them relevant candidates: (i) the presence of positively charged functional groups, which compete for electrostatic binding with viral glycoproteins; and (ii) the presence of PEGylation residues, which improves the biocompatibility of the dendrimers, thus reducing toxicity [37,38]. 

The initial phase of this study was the assessment of the cytotoxicity of dendrimers in both Vero and MRC-5 cell lines, and to do so, MTT and LDH assays were performed. Toxicity of PCCDs is a consequence of the positively charged groups in their periphery, which interacts with the negatively charged cell membrane producing nanoscale pores that affect its integrity to a point that depends on characteristics of dendrimers such as their molecular weight or the presence of fluorescent labels [39]. Masking of some of these peripheric groups was achieved by conjugation of the dendrimers with polyethylene glycol (PEG), which has previously proved to reduce immunogenicity and toxicity [40]. Results of our studies indicated that some concentrations, despite not compromising membrane integrity, had negative effects on mitochondrial activity; therefore, we selected the values from the MTT assay greater than 80% as working concentrations. No notable differences between G2-SN15-PEG, G3-SN31-PEG and G2-SN15-PEG FITC dendrimers were observed. However, G3-SN31-PEG FITC was the dendrimer presenting the highest toxicity, suggesting that larger molecular sizes and the addition of this fluorescent label enhanced the cytotoxicity of this nanostructure. 

The next step was to examine the capacity of FITC-labelled dendrimers to internalize into both cell lines by confocal microscopy and flow cytometry. Entry analyses revealed that the G2-SN15-PEG FITC dendrimer has faster uptake dynamics in both cell lines than the G3-SN31-PEG FITC dendrimer; in addition, the amount of positive fluorescent marks from both dendrimers presented an important increase along time points. Study of the fluorescent distribution showed that short times of incubation resulted in peripheral distribution, while longer times suggested that dendrimers were in more internal regions in a granular manner. This agrees with previous studies demonstrating that the size and surface charge of dendrimers influence their effectiveness of crossing cell membranes [41,42]. Our results reflect that the presence of more positive charges in the G3-SN31-PEG FITC dendrimer for being a larger molecule entailed higher affinity and stronger interactions with the cell surface, leading to longer resident times on the cell membrane, resulting in slower internalization. 

We next delved into the ability of dendrimers to inhibit HSV-2 and HCMV infections. Antiviral activity studies revealed that both studied PCCDs can inhibit these infections, with better inhibition performance than the actual reference treatments Acyclovir and Ganciclovir. In addition, we investigated the mechanism through which inhibition was achieved and results demonstrate that both dendrimers can significantly inhibit viral binding to Vero and MRC-5 cell lines; however, they are only able to directly neutralize HCMV itself and with lower inhibition values. All these results together indicate that dendrimers perform inhibition at the first stages of the infection. The fact that the G3-SN31-PEG FITC dendrimer, which has longer resident times on the cell membrane, shows better inhibition results supports this idea that inhibition is achieved by impediment of viral attachment to the cell membrane, thereby preventing further infection. 

To determine how the impediment of viral attachment takes place, we performed different analysis of the interactions of PCCDs with the receptors involved in the first stages of infection of viruses from the *Herpesviridae* family, namely the HS chains of HSPGs [36]. To do so, we started with a cell culture-based heparin competition assay, followed by 2M NaCl washing, heparinase II treatment, and “cell free” surface plasmon resonance. The first assay is based on the fact that heparin is a GAG with a similar structure to HS, thus likely able to act as an antagonist of HSPGs present on cell membranes for the binding to cationic dendrimers, leading to the observed absence of cell-associated PCCDs in both cell lines [30]. In addition, the disruption of this dendrimer-cell membrane interaction was also noticed both in Vero and MRC-5 cell lines when washing with 2.0 M NaCl, a treatment known to remove cationic molecules from cell membrane HSPGs [27]. Furthermore, digestion of the GAG moiety of HSPGs by heparinase II also lead to a notable decrease of the capacity of PCCDs to bind to cells, determined by the shift in fluorescence intensity of dendrimers in both cell lines. However, it is important to mention that there might be alternative binding sites for the dendrimers which would explain the residual amounts of dendrimers associated to cells after the performed treatments. On the other hand, SPR analysis demonstrated that these dendrimers are effectively able to bind surface-immobilized heparin (here used as a surrogate of cell-associated HSPGs) in a stable and specific way, hence with a relatively high affinity.

## 5. Conclusions

In conclusion, we can state that G2-SN15-PEG and G3-SN31-PEG dendrimers are promising therapies to face HSV-2 and HCMV infections. Biocompatibility and entry studies confirmed the suitability of these dendrimers to be used in target cells of these infections. Binding assays confirm that the studied dendrimers effectively compete with HSV-2 and HCMV for adsorption on the receptors involved in the first stages of infection of susceptible cells, leading to the impediment of viral attachment and thus producing the inhibition of these viral infections.

## Figures and Tables

**Figure 1 pharmaceutics-14-00536-f001:**
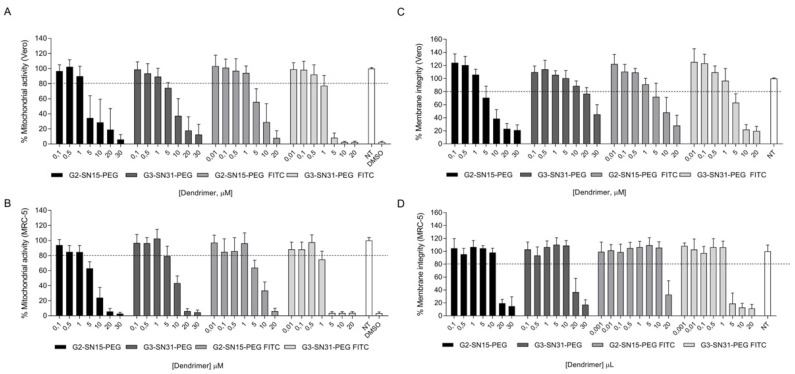
Cytotoxicity of dendrimers by 3-(4-5-dimethylthiazol-2-yl)-2,5-diphenyltetrazolium bromide (MTT) and lactate dehydrogenase (LDH) assay. (**A**,**B**) MTT assay; (**C**,**D**) LDH assay. (**A**,**C**) Vero and (**B**,**D**) MRC-5 cells were treated with increasing concentrations of dendrimers G2-SN15-PEG, G3-SN31-PEG, G2-SN15-PEG fluorescein isothiocyanate (FITC) labeled and G3-SN31-PEG FITC from 0.01 to 30 µM. Non-toxic concentrations were established when cell viability was ≥80%. Culture medium was used as cell viability control and DMSO 10% was used as death control. Data are represented as mean ± SD of three individual experiments performed in triplicate. NT: non-treated; DMSO: dimethyl sulfoxide.

**Figure 2 pharmaceutics-14-00536-f002:**
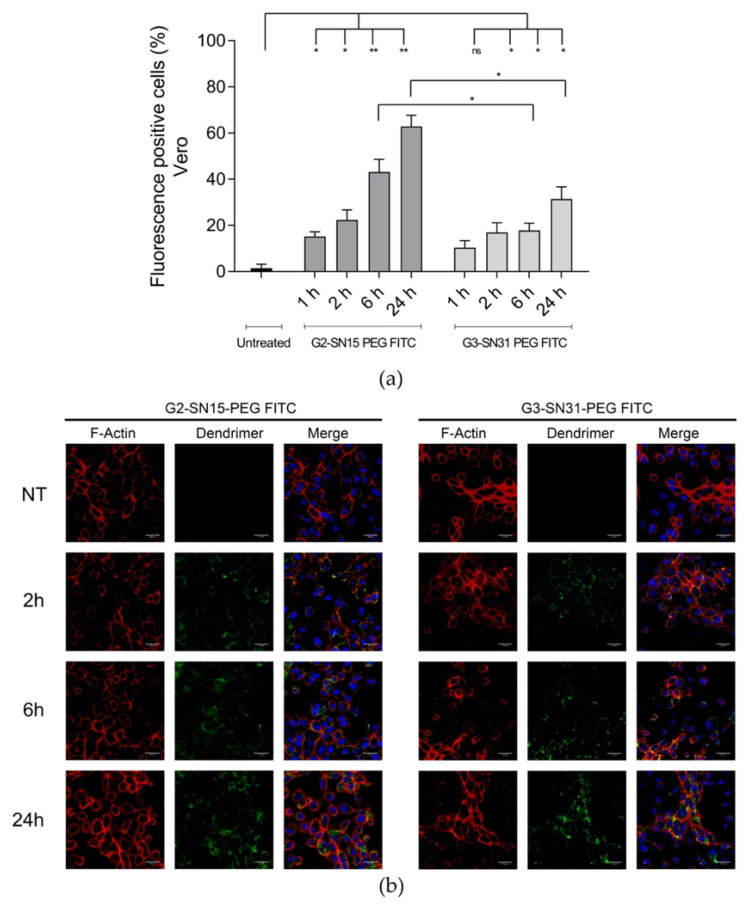
Internalization study of cationic dendrimers into Vero cells. (**a**) Percentage of FITC-positive cells observed by flow cytometry 1 h, 2 h, 6 h and 24 h post-treatment. Data are represented as mean ± SD of two individual experiments. (**b**) Representative confocal images of cells incubated with 1 µM G2-SN15-PEG FITC and 0.5 µM G3-SN31-PEG FITC dendrimers (green) for 2 h, 6 h or 24 h. Phalloidin was used to mark actin filaments (red) and DAPI for nucleus (blue). Scale bars indicate a length of 25 µm. (* *p* < 0.05; ** *p* < 0.01; ns: non-significant). DAPI: 4′,6-diamidino-2-phenylindole dihydrochloride.

**Figure 3 pharmaceutics-14-00536-f003:**
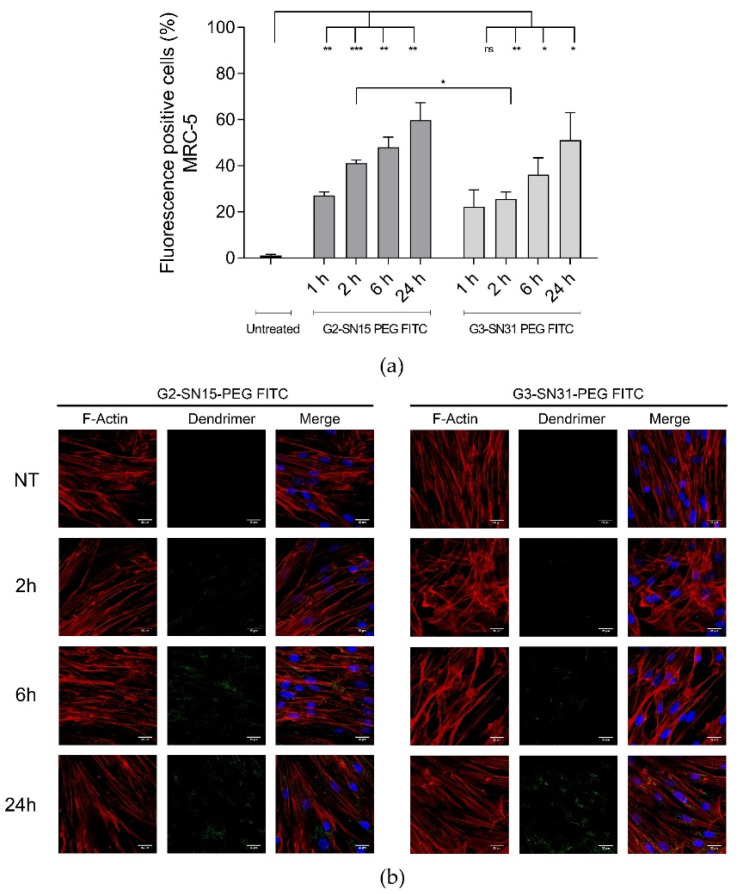
Internalization study of cationic dendrimers into MRC-5 cells. (**a**) Percentage of FITC-positive cells observed by flow cytometry 1 h, 2 h, 6 h and 24 h post-treatment. Data are represented as mean ± SD of two individual experiments. (**b**) Representative confocal images of cells incubated with 1 µM G2-SN15-PEG FITC and 0.5 µM G3-SN31-PEG FITC dendrimers (green) for 2 h, 6 h or 24 h. Phalloidin was used to mark actin filaments (red) and DAPI for nucleus (blue). Scale bars indicate a length of 25 µm. (* *p* < 0.05; ** *p* < 0.01; *** *p* < 0.001; ns: non-significant). DAPI: 4′,6-diamidino-2-phenylindole dihydrochloride.

**Figure 4 pharmaceutics-14-00536-f004:**
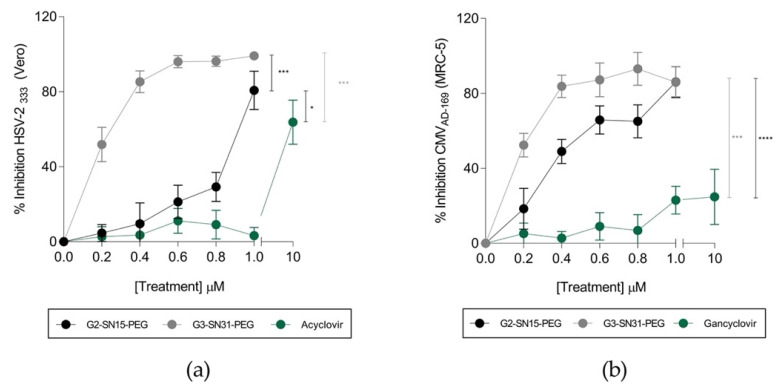
Antiviral activity of cationic dendrimers. Inhibition of herpes simplex virus 2 (HSV-2) (**a**) and human cytomegalovirus (HCMV) (**b**) infections by increasing concentrations of G2-SN15-PEG and G3-SN31-PEG dendrimers in Vero and MRC-5 cell lines, respectively. Acyclovir or Ganciclovir were used for comparison with reference treatments. Infection was measured by plaque reduction assay. Data are represented as mean ± SD of three individual experiments performed in triplicate (* *p* < 0.05; *** *p* < 0.001; **** *p* < 0.0001).

**Figure 5 pharmaceutics-14-00536-f005:**
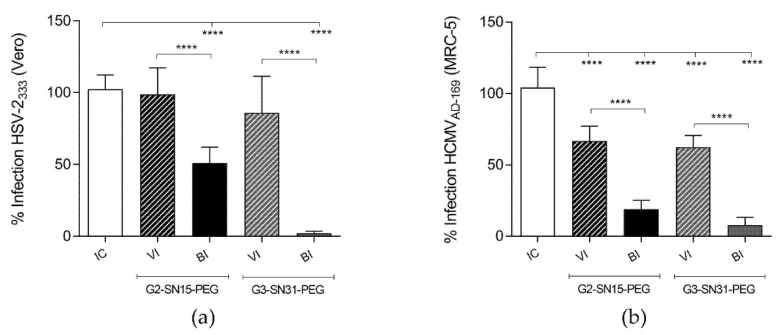
Antiviral mechanism of PEGylated cationic dendrimers. Effect of cationic dendrimers G2-SN15-PEG and G3-SN31-PEG at the maximum non-toxic concentration on HSV-2 (**a**) and HCMV (**b**) infectivity and binding. Viral inactivation (VI) and binding inhibition (BI) assays were performed, and results were measured by plaque reduction assay. Data are represented as mean ± SD of three individual experiments performed in triplicate **** *p* < 0.001.

**Figure 6 pharmaceutics-14-00536-f006:**
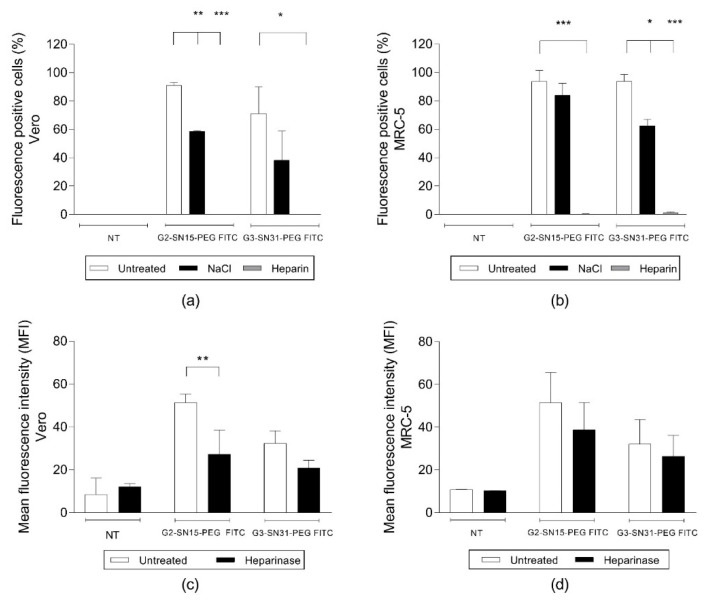
Interaction of cationic dendrimers with cell surface heparan sulphate proteoglycans (HSPGs). Fluorescent positive cells and mean fluorescence intensity of G2-SN15-PEG FITC and G3-SN31-PEG FITC dendrimers measured by flow cytometry in Vero (**a**,**c**) and MRC-5 (**b**,**d**) cells. Heparin competition assay, NaCl wash, and heparinase II treatment experiments were performed to determine the interaction of PEGylated cationic carbosilane dendrimers (PCCDs) with cell surface HSPGs. Data are represented as mean ± SD of two individual experiments * *p* < 0.05; ** *p* < 0.01; *** *p* < 0.001.

**Figure 7 pharmaceutics-14-00536-f007:**
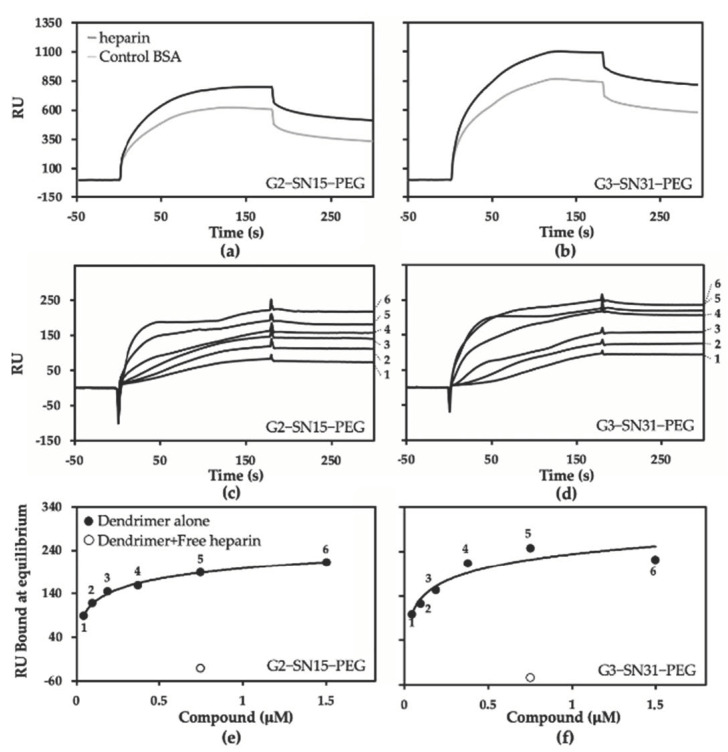
Surface plasmon resonance (SPR) analysis of G2-SN15-PEG- and G3-SN31-PEG-heparin interaction. (**a**,**b**) Sensorgrams showing the binding of G2-SN15-PEG and G3-SN31-PEG dendrimers (0.75 µM) to the heparin- or BSA-coated flow cells. (**c**,**d**) Blank-subtracted sensorgram overlays showing the specific binding of increasing concentrations of the dendrimers (1: 0.047 µM; 2: 0.094 µM; 3: 0.188 µM; 4: 0.375 µM; 5: 0.75 µM; 6: 1.5 µM) to surface-immobilized heparin. The response (in resonance units, RU) was recorded as a function of time. (**e**,**f**) Saturation curves obtained using the values of RU bound at equilibrium from panels (**c**,**d**). Result of the competition assay with free heparin is also reported.

**Table 1 pharmaceutics-14-00536-t001:** Characterization of PEGylated cationic carbosilane dendrimers.

Nomenclature	Molecular Formula	Functional Groups	Molecular Weight (g/mol)
G2-SN15-PEG	C_190_H_438_I_15_N_14_O_17_S_16_Si_13_	NMe_3_ and PEG	5987.31
G3-SN31-PEG	C_420_H_969_I_31_N_31_O_44_S_32_Si_29_	NMe_3_ and PEG	12,926.92
G2-SN15-PEG FITC	C_209_H_445_I_14_N_16_O_22_S_17_Si_13_	NMe_3_, PEG and FITC	6221.74
G3-SN31-PEG FITC	C_439_H_969_I_30_N_32_O_49_S_33_Si_29_	NMe_3_, PEG and FITC	13,161.34

**Table 2 pharmaceutics-14-00536-t002:** Kd values of the interactions of G2-SN15-PEG and G3-SN31-PEG dendrimers to heparin. Data corresponds to mean ± SEM of three individual experiments.

Dendrimer	Kd (nM) at Equilibrium
G2-SN15-PEG	368.3 ± 108.1
G3-SN31-PEG	186.7 ± 82.6

## Data Availability

Not applicable.

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
