# Peer review of "Prevention of Herpesviridae Infections by Cationic PEGylated Carbosilane Dendrimers"

_pharmaceutics, 2022, doi:10.3390/pharmaceutics14030536_

Round 1
Reviewer 1 Report
The study has good results on infections analysis but the characterization of their formulation is insufficient. Thus, the paper can be accepted after addressing the following comments:
1-why the authors chose dendrimer as the formulation. What is its specific feature in the antiviral application?
2-how do you sure about synthesize dendrimers? There is no test to validate that.
3-what about the morphology of your formulations?
- the resolution of Figures 1 and 7 is low. Please consider
5- it is better to consider some new papers in this field and discuss them in the introduction. For example, added some other nanocarriers such as liposomes and niosomes for infection diseases
Author Response
- Why the authors chose dendrimer as the formulation. What is its specific feature in the antiviral application?
Nanotechnology is a promising strategy for targeting viral infections. There are different formulations that have proved to be useful to face viral infections, among them dendrimers have proved themselves to be potent and safe antiviral agents. They stand out for being well-defined polymers with useful functional groups at the periphery, a reasonable manufacture cost and a controlled synthesis. In addition, they are highly biocompatible, soluble, polyvalent, and present low polydispersity. All these characteristics differentiate them form other polymers and make them suitable for therapeutic approaches such as inhibition of viral infections [1]. We have added this information to the introduction to highlight the usefulness of this type of molecules.
- How do you sure about synthesize dendrimers? There is no test to validate that.
Dendrimers are monodispersed macromolecules with a uniform architecture. They are characterized by an easy, defined and precisely designed structure [2].
Authors would like to mention the fact that there is a complementary paper under revision entitled “Promising cationic PEGylated carbosilane dendrimer as new antiviral agents: synthesis and preliminary anti-HIV-1 and anti-HSV-2 activity” which is focused on the synthesis of these dendrimers and details different specific aspects and tests made to validate their formulation.
- What about the morphology of your formulations?
The morphology of the formulations of the dendrimers used in this work can be found in Figure 8 of the paper cited in the text: Royo-Rubio, E.; Rodriguez-Izquierdo, I.; Moreno-Domene, M.; Lozano-Cruz, T.; de la Mata, F.J.; Gomez, R.; Munoz-Fernandez, M.A.; Jimenez, J.L. Promising PEGylated cationic dendrimers for delivery of miRNAs as a possible therapy against HIV-1 infection. J Nanobiotechnology 2021, 19, 158, doi:10.1186/s12951-021-00899-0.
- The resolution of Figures 1 and 7 is low. Please consider.
We agree with the observation that Reviewer has made about the quality of Figures 1 and 7. We have improved the resolution of both figures.
- It is better to consider some new papers in this field and discuss them in the introduction. For example, added some other nanocarriers such as liposomes and niosomes for infection diseases.
We agree with the observation that Reviewer has made related to the appropriateness to add more recent papers in this field in the introduction. We have added various references related to different nanoparticles used as antiviral agents.
Reviewer 2 Report
Authors report “Prevention of Herpesviridae infections by cationic PEGylated 2 carbosilane dendrimers” Manuscript could be interesting for readers. However, it needs modifications as suggested below:
Comments
1 Abstract should be revised and briefly results should be incorporated.
2 It should be justified why cytotoxicity is evaluated over a period of 48 h and 6 days?
3 There should be space between the number and unit.
4 Resolution and clarity of Figure 1 a-d and Figure 2a should be improved.
5 Figure 4, clear and colorful graphs should be included.
6 Result and discussion should be elaborated with relevant citations
7 Page no.8. It should be explained, why G2-SN15-PEG FITC dendrimer has faster uptake dynamics than G3-SN31-PEG FITC dendrimer and how cellular uptake by dendrimer can be determined?
8 What is the size of dendrimers used in this study?
9 Conclusion should be revised and further elaborated briefly.
10 What is the impact of this study?
11 Typographical and grammatical mistakes should be corrected.
12 Some more current references should be cited.
Author Response
Reviewer 2:
- The abstract should be revised and briefly results should be incorporated.
We agree with the observation that Reviewer has made about the neediness to incorporate results in the abstract. This part has been revised and changed.
- It should be justified why cytotoxicity is evaluated over a period of 48 h and 6 days?
The cytotoxicity is evaluated over a period of 48 h in the Vero cell line and 6 days in the MRC-5 cell line because that is how long inhibition experiments take. The authors have added this clarification in the methodology for better understanding.
- There should be space between the number and unit.
The authors would like to thank the reviewer for this appreciation, this error has been corrected and was found.
- Resolution and clarity of Figure 1 a-d and Figure 2a should be improved.
We agree with the observation that the Reviewer has made about the quality of Figures 1 (a-d) and 2a. We have improved the resolution and clarity of both figures.
- Figure 4, clear and colorful graphs should be included.
The authors have changed Figure 4 to clarify the information given in these graphs.
- Result and discussion should be elaborated with relevant citations
References used to elaborate results and discussion have been changed to introduce more recent and relevant information.
- Page no.8. It should be explained, why G2-SN15-PEG FITC dendrimer has faster uptake dynamics than G3-SN31-PEG FITC dendrimer and how cellular uptake by dendrimer can be determined?
Dendrimer uptake into target cells differs depending on dendrimer properties like size or surface charge and depending on cell characteristics as membrane chemical composition [3,4]. These characteristics will determine the effectiveness of the bound of the dendrimer to the cell surface: more positive charges in the periphery entail stronger interactions with cell membrane leading to slower internalization. This has been previously observed with these same dendrimers in different cell lines (PBMCs and U87MG-CD4+CCR5+ cells) [5].
The determination of this uptake has been done by measuring the number of positive cells for dendrimers by flow cytometry and by confocal microscopy visualization. This has been included in the discussion.
- What is the size of dendrimers used in this study?
The hydrodynamic diameter of the dendrimers used in this study can be found in Table 3 of the cited paper: Royo-Rubio, E.; Rodriguez-Izquierdo, I.; Moreno-Domene, M.; Lozano-Cruz, T.; de la Mata, F.J.; Gomez, R.; Munoz-Fernandez, M.A.; Jimenez, J.L. Promising PEGylated cationic dendrimers for delivery of miRNAs as a possible therapy against HIV-1 infection. J Nanobiotechnology 2021, 19, 158, doi:10.1186/s12951-021-00899-0.
For easier referencing, G2-SN15-PEG dendrimer is 302.95 ± 14.78 nm and G3-SN31-PEG dendrimer is 4.93 ± 0.26 nm.
- The conclusion should be revised and further elaborated briefly.
We agree with the observation that the Reviewer has made related to the neediness to further elaborate the conclusions, we have made changes to this part to improve it.
- What is the impact of this study?
The objective of this work was to study a new promising therapy against Herpesviridae infections, which are among the most widespread pathogens in the world but still have no cure due to characteristics of these infections such as the appearance of resistance to current treatments.
Our work has demonstrated the potential of two PEGylated cationic dendrimers (G2-SN15-PEG and G3-SN31-PEG) against HSV-2 and HCMV, which have been proved to inhibit the interaction between viral glycoproteins and cell HSPGs. The prevention of this interaction entails a significant inhibition of the infection, which means that they could be used as a new therapy to face these infections.
- Typographical and grammatical mistakes should be corrected.
We would like to thank the Reviewer for the appreciation of the presence of typographical and grammatical mistakes in the text. We have corrected them where found.
- Some more current references should be cited.
We agree with the observation that the Reviewer has made related to the appropriateness to add more recent papers in this field. We have changed our previous references to more recent ones in all parts of the manuscript.
We would also like to mention that changes have been made to improve the English language. In addition, we have made changes to improve the information given in the introduction and the description of the conclusions.
Reviewer 3 Report
The manuscript titled, "Prevention of Herpesviridae infections by cationic PEGylated carbosilane dendrimers" by Elena et al., is a comprehensive study of action of PEGylated dendrimers on Herpesviridae family of viruses. The manuscript is well written by discussing the problems that we face on developing antiviral agents and then the introduction of designed dendrimers used in this study.
- The axes label of Figure 1 can be improved to improve visibility.
- Non toxic concentrations for SN15 and SN31 remains the same except at 0.5 µM concentration of SN31-FITC. How the authors explain or think about this difference. Is it because FITC also helps to kill viral cells.
- I would also like the authors to include a simple cartoon to explain how these dendrimers bring about viral inhibition or binding to facilitate the complete understanding of this phenomenon.
Author Response
- The axes label of Figure 1 can be improved to improve visibility.
We agree with the observation that the Reviewer has made about the quality of Figure 1, we have improved the resolution of the figure.
- Nontoxic concentrations for SN15 and SN31 remain the same except at 0.5 µM concentration of SN31-FITC. How the authors explain or think about this difference? Is it because FITC also helps to kill viral cells.
The fact that G2-SN15-PEG and G3-SN31-PEG present the same toxicity but G3-SN31-PEG FITC presents higher toxicity is due to the presence of the fluorophore (as has been demonstrated in previous works), which increases the toxicity of the nanocompound more than the increase in the generation of the dendrimer.
- I would also like the authors to include a simple cartoon to explain how these dendrimers bring about viral inhibition or binding to facilitate the complete understanding of this phenomenon.
The authors would like to ask Reviewer 3 if the cartoon mentioned should be included as a new figure in the manuscript or as a graphical abstract.
Round 2
Reviewer 1 Report
The paper can be accepted in this format